# Comparative RNA-Seq Analysis Revealed Tissue-Specific Splicing Variations during the Generation of the PDX Model

**DOI:** 10.3390/ijms242317001

**Published:** 2023-11-30

**Authors:** Eun Ji Lee, Seung-Jae Noh, Huiseon Choi, Min Woo Kim, Su Jin Kim, Yeon Ah Seo, Ji Eun Jeong, Inkyung Shin, Jong-Seok Kim, Jong-Kwon Choi, Dae-Yeon Cho, Suhwan Chang

**Affiliations:** 1Department of Physiology, University of Ulsan College of Medicine, Asan Medical Center, Seoul 05505, Republic of Korea; ej00230@amc.seoul.kr (E.J.L.); sim1504@naver.com (M.W.K.); kkkk9462@naver.com (S.J.K.); seoyeonahh98@gmail.com (Y.A.S.); jieun6570@naver.com (J.E.J.); 2PentaMedix Co., Ltd., Seongnam 13449, Republic of Korea; sjnoh@pentamedix.com (S.-J.N.); choihs@pentamedix.com (H.C.); dabb@pentamedix.com (D.-Y.C.); 3Myunggok Medical Research Institute, College of Medicine, Konyang University, Daejeon 35365, Republic of Korea; jskim7488@konyang.ac.kr (J.-S.K.); jabuss@naver.com (J.-K.C.)

**Keywords:** alternative splicing, PDX model, tissue specificity, RNA-seq

## Abstract

Tissue-specific gene expression generates fundamental differences in the function of each tissue and affects the characteristics of the tumors that are created as a result. However, it is unclear how much the tissue specificity is conserved during grafting of the primary tumor into an immune-compromised mouse model. Here, we performed a comparative RNA-seq analysis of four different primary-patient derived xenograft (PDX) tumors. The analysis revealed a conserved RNA biotype distribution of primary−PDX pairs, as revealed by previous works. Interestingly, we detected significant changes in the splicing pattern of PDX, which was mainly comprised of skipped exons. This was confirmed by splicing variant-specific RT-PCR analysis. On the other hand, the correlation analysis for the tissue-specific genes indicated overall strong positive correlations between the primary and PDX tumor pairs, with the exception of gastric cancer cases, which showed an inverse correlation. These data propose a tissue-specific change in splicing events during PDX formation as a variable factor that affects primary−PDX integrity.

## 1. Introduction

The patient-derived xenograft (PDX) model is a valuable resource for studying the biology of human tumors and examining the efficacy of various therapeutic agents [1]. Most reports, by far, have shown that the molecular nature of PDX compared to the primary tumor does not deviate much, supporting this model as a represent to human counterpart [2,3]. However, certain reports have also demonstrated molecular changes in the PDX tumor that seem to be dependent on the individual character or cancer type [4]. Considering the dramatic environmental changes during xenograft, a selection process among the heterogenous primary tumor cells likely to occur in order to fit into the animal model [5]. Despite this intrinsic limitation, the PDX model is widely used and is considered to be a preclinical tool that strongly reflects human tumors. Many research groups, including our team, have generated PDX models from various types of tumor [2,3]. Depending on the tumor type and its stage, the success rate of PDX formation varies. In particular, if the route of the initial graft is subcutaneous, then the microenvironment, including the tissue matrix, surrounding cell type, and vasculature, can differ significantly from the original site of the tumor. So, it is possible that cancer cells adopt various molecular mechanisms of gene expression regulation in order to adjust to the new environment.

Tissue-specific gene expression has been studied extensively as it is fundamental for explaining the uniqueness of each tissue [6]. Following this simple idea, it is reasonable to assume that each tumor type will also present a different repertoire of gene expressions that will confer specific characteristics. In addition to genetic mutations, recent studies have highlighted the impact of alternative splicing events that lead to tumor diversity and plasticity [7]. In addition, alternative splicing controls metastatic nature [8], chemoresistance [9], and prognosis [10]. Therefore, it is important to consider the cancer-specific splicing event to understand its biology and find suitable therapeutic agents [11]. Also, it is meaningful to analyze how splicing events are affected during the formation of PDX tumors. Related to this idea, a previous study performed a comprehensive, proteogenomic analysis for triple negative breast cancer including splicing variation analysis [12]. Several pan-cancer analyses indicated that tumor-type-specific splicing events [13,14] provide a useful tool for classification [15] and therapeutic application [16]. Importantly, understanding which splicing variations occur in a tissue-specific manner (or tumor type dependent manner) during PDX formation will provide us useful insight when using the PDX model. Based on these ideas, we analyzed four different primary−PDX tumor pairs using RNA-seq and the results are presented here.

## 2. Results

### 2.1. Generation of PDX Model from Various Types of Primary Tumors

To obtain the PDX model from various tissues, we used 20 xenografts from gastric, esophagus, sarcoma, colon, breast, lung, and head and neck cancers. We were able to establish 6 PDX models out of 20 (Success rate 30%), including breast, stomach, lung, and colon cancer (Appendix A). An image of the PDX tumor and its growth curves are shown in Appendix A and Figure 1, respectively. Compared with gastric and breast PDX, we observed that lung and colon PDX grew slower.

### 2.2. RNA-Seq Analysis of Four Primary−PDX Pairs Shows Conserved RNA Biotype Distribution with Significant Correlation for the Expression of Tissue-Specific Genes

We next questioned where there were expression variations between primary and PDX tumors in a tissue-specific manner. For this, we performed an RNA-seq analysis of four pairs of primary−PDX from gastric (KY009), lung (KY011), breast (KY013), and colon (KY018) cancers. The overall analytic pipeline is described in Section 3. As previously reported, the RNA biotype distribution of the four pairs of primary−PDX tumors was well conserved (Appendix A), with a reduced portion of antisense transcript level in the case of gastric primary tumor (Appendix A). The DEG (differentially expressed genes) analysis for each primary−PDX pair showed a varying number of genes, with highest number being in gastric cancer (Appendix A). The top 10 differentially expressed genes in the primary tumor/PDX pairs are summarized in Appendix A. When we focused on tissue-specific genes, we found variable numbers of differentially expressed genes for each of the primary−PDX pairs (Figure 2A). Interestingly, most of the tissue-specific genes were highly correlated in the pairs, indicating molecular conservation of the tissue-specific genes during PDX formation (Figure 2B–E). However, we found the gastric-specific genes were inversely correlated in the primary−PDX tumors (see Section 4), suggesting unique molecular changes in gastric PDX formation.

The pathway analysis of the differentially expressed genes showed significantly enriched immune cell activation and cytokine/chemokine signaling in stomach PDX (KY009), whereas breast (KY013) and colon (KY018) PDX showed downregulation for such pathways (Figure 3A–D). Except for KY009, PDX did not show an enriched pattern for upregulated pathways, with a low number of upregulated genes (Appendix A). Rather, they were enriched for downregulated pathways, and protein processing (degradation) and antigen presentation were commonly found (Figure 3E–H).

### 2.3. RNA-Seq Analysis of Four Primary−PDX Pairs Revealed a Large Number of Splicing Variations in a Tumor-Type-Specific Manner

In addition to tissue-specific gene expression changes, alternative splicing is a major cause for transcript diversity in tumors. Hence, we next analyzed the primary−PDX RNA-seq data to seek splicing changes during PDX formation and to find tissue-specific alternative transcripts. Figure 4A shows the total differential splicing events detected in each PDX formation, indicating that thousands of splicing events are changed during xenograft formation. The top 10 differentially spliced genes in primary tumor/PDX pairs are summarized in Appendix A. Splicing pattern analysis revealed that skipped exons (SEs) comprised 45% (gastric) to 63% (breast) of all splicing events (Figure 4B). The number and percent of each splicing type were also analyzed when low significance (FDR > 0.05) events were included (Appendix A). Volcano plots for each primary−PDX pairs showed a significant number of upregulated or downregulated splicing events, and their patterns varied by tumor type. While gastric and colon PDX showed more downregulated events, lung PDX contained more upregulated events (Figure 4C–F).

To gain functional insights for these splicing variations, we further analyzed these variations for Gene Ontology and Pathways. As shown in Figure 5, we found unique enriched pathways or molecular functions in each type of PDX, implying that the differential splicing events functioned in a tissue-specific manner. Despite this, we found commonly upregulated cell adhesion and cytoskeleton genes, implying cancer cells utilize alternative splicing to adjust to rapid changes in the external environment induced by xenografts.

### 2.4. Validation of the Skipped Exon Pattern for Each Primary−PDX Pair by RT-PCR

To validate the alternative splicing pattern discovered from the RNA-seq analysis, we performed RT-PCR for the skipped exon using primers binding to the adjacent exons. Because the top splicing events (determined by the difference between primary and PDX) were varied by each tumor type (Appendix A), we designed four primer sets amplifying the top splicing genes for each tumor type (Appendix A). The sequences are provided in Appendix A. For the exon skipping event, the number of larger fragments were expected to be decreased (or smaller PCR product size to be increased). Indeed, Figure 6A shows a smaller splicing product of *SLK* and *CTNND1* that increased in the PDX lane (*RPL13a* was used as a control). Likewise, we observed splicing variation in the KY011 tumor for *COASY* (Figure 6B). In contrast, splicing analysis of *SERPINA* showed a reduced large-sized transcript in the KY013 PDX tumor (arrow), while the smaller band remained the same (asterisk, Figure 6C, upper panel). A similar difference was observed in the *SENP6* gene as well (Figure 6C, mid panel). In the KY018 tumor, we found altered splicing in the *APOPT1* gene, with a relatively stronger signal in the lower band in the PDX tumor (Figure 6D). We also examined two other genes, *EPB41L2* and *FBLN2*, that were previously reported to be differentially spliced [14]. The results in Figure 6D show an increased upper band of *EPB41L1* in the KY013 pair, while the upper band of *FBLN2* was decreased for the KY018 pair. In the case of *CASK*, we observed multiple bands that were increased in PDX (Appendix A, see discussion). For *CD44*, we could not find an alternative splicing product (Appendix A). These results collectively demonstrate dynamic splicing alterations during PDX formation, and the events differed by tumor type.

## 3. Materials and Methods

### 3.1. Generation of Patient-Derived Xenograft (PDX) Models

All of the animal studies were conducted at the Asan Institute for Life Sciences (Seoul, Republic of Korea) and were approved by the International Animal Care and Use Committee (IACUC). BALB/c female mice aged 5 to 6 weeks were supplied by JA Bio and used in the experiments. Fresh tumor tissues were collected (or obtained) from human patients at the Konyang University Hospital. The tissues were placed into tubes containing RPMI medium and then packaged in an icebox. These were sent to the Asan Medical Center (Seoul) as quickly as possible. The samples were washed with RPMI medium and cut into small pieces of 2~3 mm^3^. Needle biopsy tissues that were too small were cut once or used as is. Tumor fragments were implanted subcutaneously into the flanks (for gastric, lung, and rectal cancers) and subcutaneously into the inguinal mammary fat pads (for breast cancers) of 6- to 14-week-old mice. The mice were anesthetized with 2~3% isoflurane (Terrel isoflurane, USA) for induction and 1.5–2.0% maintenance. Tumor growth was monitored from 2 weeks to 6 months after implantation. When the tumor volume reached 400~1000 mm^3^, the mice were euthanized and the tumors were harvested. A portion of the tumors were reimplanted for passage and the remaining amount was stored as a stock in a freezing medium (FBS 9: DMSO 1). H&E staining was performed on part of the remaining tissues for the histological analysis.

### 3.2. RNA-Seq Analysis

More than 30 million reads were generated for each of the 4 primary tumor tissues and 4 PDX samples with a median of 48.6 million reads. After discarding the adapter sequences and trimming the low-quality reads, the reads were aligned to the hg19 reference genome using the STAR aligner (v2.7.3) with default settings. Aligned reads were then sorted using SAMtools (v1.10) and duplicate reads were removed using MarkDuplicates (GATK v4.1.4.0). The aligned reads were then counted using HTseq (v0.11.3), which evaluated the expression levels of the transcripts overlapping their exons for each gene. The NOIseq R package (v2.42.0) [17] was utilized as it is explicitly intended to detect differentially expressed genes (DEGs) for RNA-Seq data, even in situations wherein there are no replicates available. The read count data were assessed using the biodetection, rpkm, and noiseq functions within the NOIseq R package. The RNA biotype distribution was determined through biodetection, and RPKM normalization was employed before identifying differentially expressed genes (DEGs) between PDX samples and their corresponding primary tumors, using default options. Next, the ‘clusterProfiler’ R package (v4.6.2) [18] was utilized to examine the biological functions and pathways related to the DEGs. The package incorporates gene ontology databases like GO (https://geneontology.org/docs/download-ontology/, accessed on 11 September 2023) and KEGG (https://www.kegg.jp/kegg/rest/keggapi.html, accessed on 11 September 2023). To ascertain any discrepancies in tissue-specific gene expression between PDXs and primary samples, Pearson correlation analysis was performed. To facilitate this, we procured tissue-specific gene lists from publicly available databases, including TiGER (http://bioinfo.wilmer.jhu.edu/tiger/download/hs2tissue-Table.txt, accessed on 5 September 2023) [19] and TissGDB (https://bioinfo.uth.edu/TissGDB/download_dir/g_ctype_tissue.txt?csrt=12242179511302922869, accessed on 5 September 2023) [20]. TissGDB contains 40, 116, 140, and 208 tissue-specific genes for breast, colon, stomach, and lung cancer, respectively. Meanwhile, TiGER has 82, 148, 142, and 88 genes for the same tissue types. Among them, 3, 66, 57, and 37 genes were common to both TiGER and TissGDB. Finally, differential alternative splicing events between primary tissues and PDXs were analyzed using rMATS software (v4.1.2) [21]. We filtered for significant differential splicing events with FDR < 0.05 and IncLevelDifference > 0. Gene ontology analysis was also performed on the genes presenting differential splicing events.

### 3.3. PCR-Gel Electrophoresis for Validation of the Splicing Events

For validation of the splicing events, we used three pairs of primary−PDX RNAs. The total RNA extraction was performed using TRizol (Invitrogen, Waltham, MA, USA), according to the manufacturer instructions. cDNA synthesis was performed using the PrimeScript™ RT reagent Kit (RR037A, Takara, Kusatsu, Japan) according to the manufacturer’s guidelines. We used 500 ng RNA for cDNA synthesis, and each reagent was used in the amounts indicated in the table below. Reverse Transcription PCR was performed using a T100 Thermal Cycler. PCR was performed using AccuPower^®^ PCR PreMix (K-2016) according to the manufacturer’s guidelines. The reagents were used for PCR in amounts corresponding to the table below. The template DNA from cDNA synthesis was diluted one-third to 3’ D.W. and used at 1 μl. PCR was performed using a T100 Thermal Cycler. Gel electrophoresis was performed to confirm the size of the DNA. Gel electrophoresis was performed by mixing 0.0025% Dyne Loading Star (A750, DYNE BIO, Seongnam, Republic of Korea) and Certified Molecular Biology Agarose (1613102, Bio-rad, Hercules, CA, USA) in 0.5X TAE buffer (C-9004, Bioneer, Oakland, CA, USA) and boiling to create a 1.5% agarose gel. Gel electrophoresis was performed using the i-MyRun.NC Agarose Gel Electrophoresis Set (IMR-303, Cosmo Bio, Tokyo, Japan), and the gel electrophoresis results were analyzed using the ChemiDoc MP Imaging System (12003154, Bio-rad). The densitometry analysis of the protein was calculated using ImageJ (v1. 54f). Blot images were imported into ImageJ. We selected the region of interest (ROI) and obtained the pixel densities of each band. All of the pixel densities were then calculated.

### 3.4. Statistical Analysis

Data are presented as standard error of the mean (SEM). Statistical significance was determined using the two-tailed Student’s *t*-test. For all of the measurements, *p*-values less than 0.05 were regarded as statistically significant.

## 4. Discussion

In our study, we found tumor-type-specific splicing variations in four different PDX models. Among these, the *CTNND1* (p120 catenin) isoform switch was previously shown to induce tumor cell invasion and to predict metastatic disease via Rho GTPase [22]. This event is known to be regulated by epithelial splicing factors of ESRP1 and ESRP2, which are also involved in the splicing of *CD44, FGFR2*, and *ENAH* [23]. Recently, ESRP1 was reported to regulate the isoform switch of *LRRFIP2* and to determine the metastasis of gastric cancer [24]. Hence, it seems that multiple target genes regulated by the ESRP splicing factor reshaped the primary tumor to fit into the new microenvironment of the animal model. Related to this, the alternative splicing of CD44 was reported to enhance the colonization of metastatic lung and colorectal cancer cells [25,26]. Moreover, a regulator named SFPQ was shown to be overexpressed and to promote lung cancer malignancy via CD44 v6 expression. These results suggest the regulation of CD44 splicing by multiple regulators control lung cancer aggressiveness and thereby affect the success of PDX formation. In contrast with CD44, the CASK gene is not well characterized for its alternative splicing, with the exception of insertional exon 24a (E24a), which is induced by neuronal stimulation [27]. We found multiple ladders for the RT-PCR of CASK (Figure 6C), suggesting primer slippage during PCR or another molecular event such as repetitive elements. Lastly, tissue-specific splicing variation is known for SERPINA1 in alpha-1 antitrypsin deficiency [28], but its function in cancer is unclear. A report published in 2015 presented that Snail and SERPINA1 promote tumor progression in colorectal cancer [29]. In breast cancer, SERPINA1 was shown as a direct target gene of estrogen receptors and as a predictor of survival [30], but its splicing was not studied much. Hence, it is reasonable to speculate that more alternative splicing could be found and that it should functionally be studied in the process of xenograft tumor formation. Indeed, when we tested EPB41L2 and FBLN2, which are reported cancer-type-specific spliced genes [14], we found decreased exon skipping of EPB41L2 in breast PDX, whereas the FBLN2 gene showed increased exon skipping in colon PDX (Figure 6D). Among the various splicing types, exon skipping comprised about 50% of the differential splicing event in PDX (Figure 5B). A previous report also showed a similar portion of skipped exon levels and this type seemed to increase during the evolution process [31]. Therefore, the skipped (or included) exon might be a major determinant that drives the human-specific nature of specific organs and, accordingly, the tumor characteristics. Indeed, a recent report revealed MET exon 14 skipping results in Crizotinib resistance in a lung PDX model [32].

Pathway analysis for the differentially spliced genes during PDX formation (Figure 5) indicated an upregulation in focal adhesion for KY009 (gastric), adherence junction for KY013 (breast), and cadherin binding for KY018 (colon). These were all related to the cell−cell or cell−matrix interaction, which seems to be important for the stable attachment of the grafted tumor mass onto the endogenous mouse tissue. However, similar pathway terms were also found in the downregulated part, making interpretation difficult. At present, we speculate that alternative splicing for some of the genes classified in the cell−cell or cell−matrix interaction pathways result in diverse functional changes during the adaptation of tumor cells in mouse tissue. Further study about the functional effect of splicing will clarify this point.

One of the limitations of our study was that there was one primary−PDX pair for each tumor type. A previous comprehensive study of 536 PDX models across 25 cancer types revealed dynamic genomic landscapes and pharmacogenomic associations [33]. More specifically, the authors classified PDX models according to the top 1000 most variable genes from cancer types using more than 20 samples, and this analysis identified four major transcriptional groups. These were squamous-enriched, connective-tissue-enriched, digestive system, and low expression groups. Even though they did not consider splicing events, the results suggested that the expression profile of PDX could be classified according to the origin of the cell or organ system. In our study, we analyzed four different tumor types that showed quite different expression patterns (Figure 3A) and splicing events during PDX formation (Appendix A for top differentially spliced genes). Further study using more primary−PDX tumor pairs will clarify if the differential splicing event found here was indeed conserved in each tumor type.

## Figures and Tables

**Figure 1 ijms-24-17001-f001:**
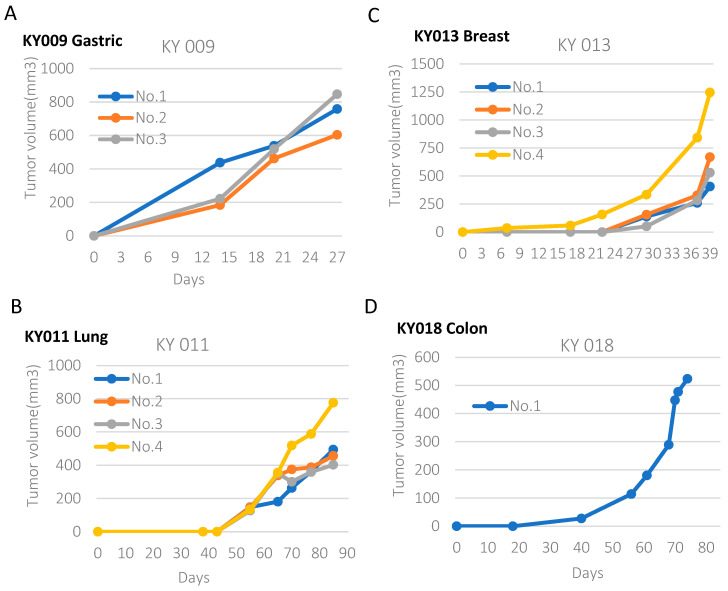
Generation of the PDX model from four tumor types (**A**–**D**). Growth curve for gastric ((**A**), KY009), lung ((**B**), KY011), breast ((**C**), KY013), and colon ((**D**), KY018) PDX xenografts. Pictures of the PDX mouse and dissected tumors are provided in the Appendix A.

**Figure 2 ijms-24-17001-f002:**
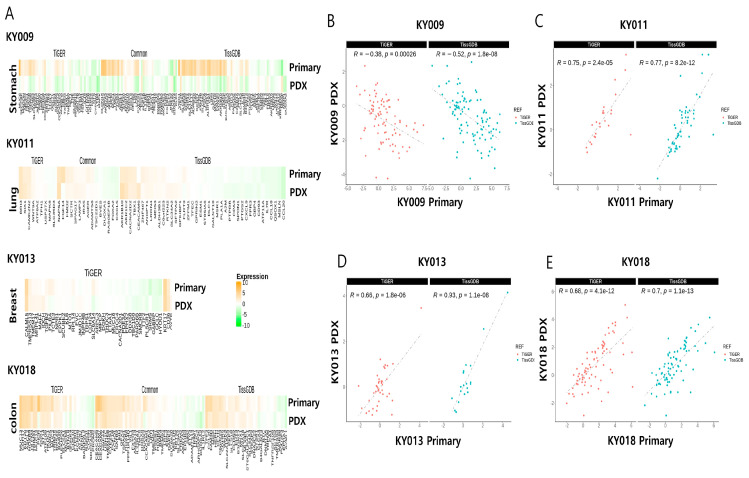
The expression of tissue-specific genes are largely conserved during PDX formation in three tumor types. (**A**) The heatmap of the tissue-specific genes (upper row, primary tumor; lower row, PDX) for KY009, KY011, KY013, and KY018, indicating a conserved expression pattern during PDX formation, with the exception of KY009 (**B**–**E**). Graphs showing Pearson’s correlation test for tissue-specific genes between primary and PDX tumor of gastric (**B**), lung (**C**), breast (**D**) and colon (**E**).

**Figure 3 ijms-24-17001-f003:**
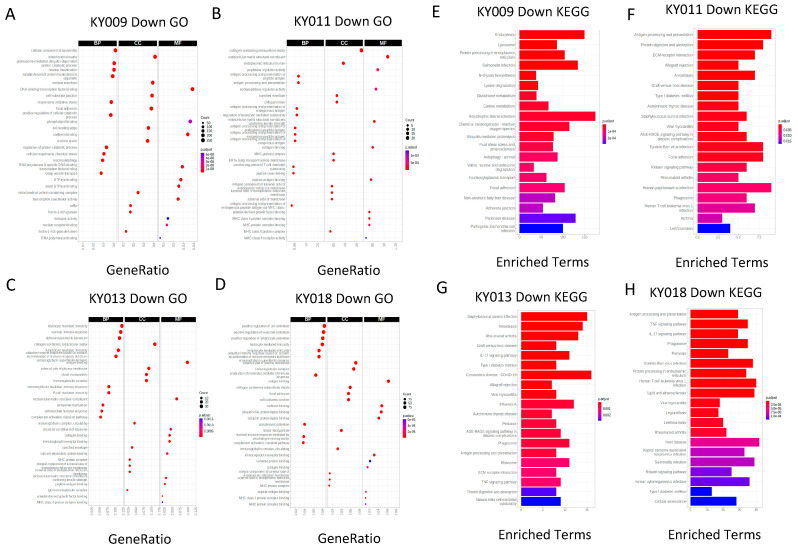
Pathway analysis showing tumor-type specificity, but antigen processing function is commonly downregulated. (**A**–**D**) Gene ontology analysis of differentially expressed genes between primary and PDX tumors of gastric (**A**), lung (**B**), breast (**C**), and colon cancer (**D**). BP, biological process; CC, cellular compartment; MF, molecular function. (**E**–**H**) KEGG analysis of the same primary−PDX tumors for top-down-regulated pathways. Molecular pathways are listed on the left and the number of enriched terms are marked on the x axis.

**Figure 4 ijms-24-17001-f004:**
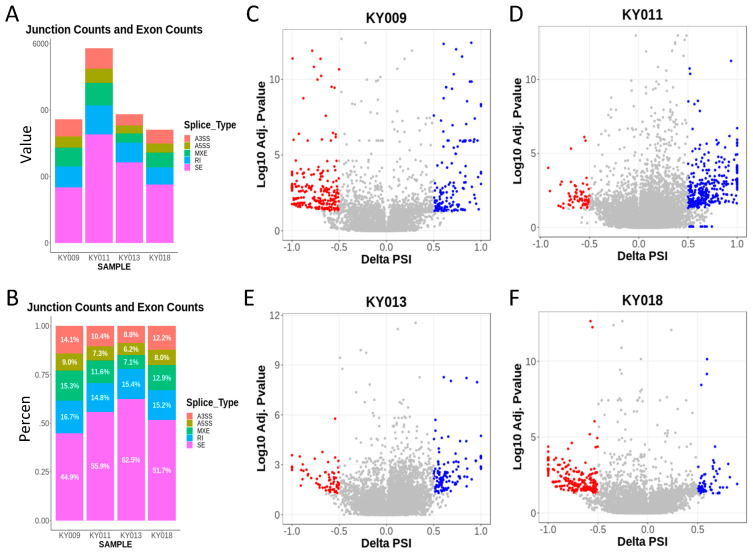
Significant numbers of alternative splicing events are present, with the highest alterations in lung cancer. (**A**,**B**) Number (**A**) and percent (**B**) of differential splicing events between primary and PDX tumors. Note: skipped exons (pink) comprised the largest portion of differential splicing. SE: skipped exon; A5SS: alternative 5’ splice site; A3SS: alternative 3’ splice site; MXE: mutually exclusive exons; RI: retained intron. (**C**–**F**) Volcano plots showing the distribution of the significantly altered splicing genes in gastric (**C**), lung (**D**), breast (**E**), and colon cancer (**F**). Blue dots indicate gene splicing upregulated in PDX, while red dots indicate gene splicing downregulated in PDX.

**Figure 5 ijms-24-17001-f005:**
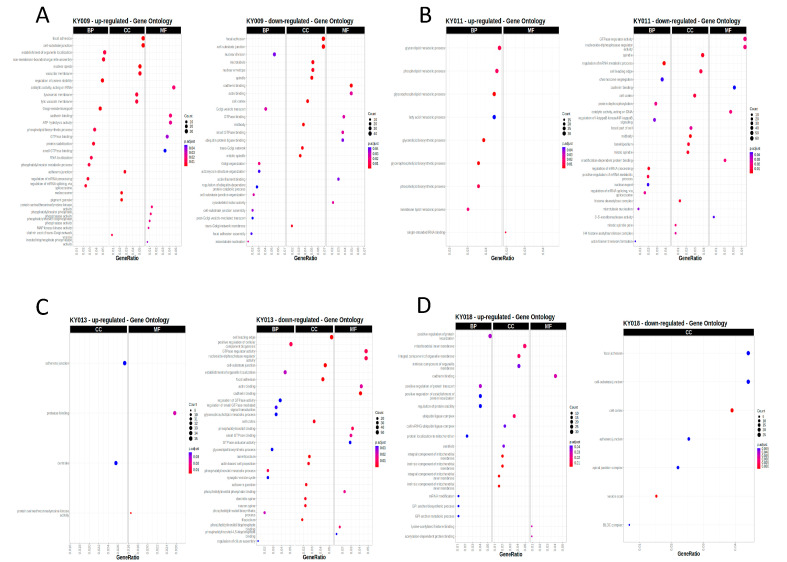
Unique enriched pathways or molecular function of the alternative splicing events for each type of tumor. (**A**–**D**) Gene Ontology analysis for differential splicing events between primary and PDX tumors of gastric (**A**), lung (**B**), breast (**C**), and colon cancer (**D**). Left panel shows upregulated pathways; right panel shows downregulated pathways. The enriched pathways are listed on the y axis and the x axis marks the gene ratio. Size of dots indicate the number of genes included in the pathway. Color of dots reflects statistical significance. BP, biological process; CC, cellular compartment; MF, molecular function.

**Figure 6 ijms-24-17001-f006:**
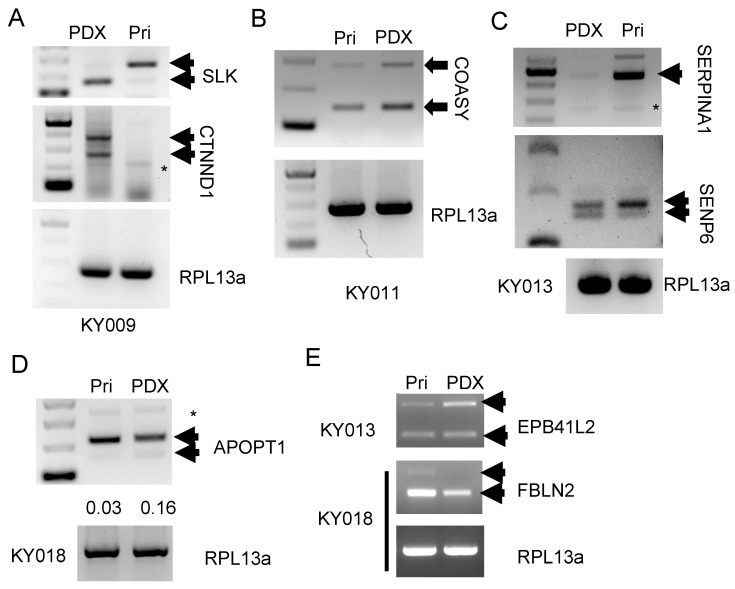
RT-PCR using exon skipping specific primer confirms the alternative splicing events during PDX formation. (**A**) Agarose gel electrophoresis of RT-PCR products for the splicing products of *SLK* and *CTNND1* genes. In the KY009 tumor. The two arrows show unspliced/spliced products. The asterisk (*) indicates a non-specific band. *RPL13a* (lower panel) is used as the internal control. (**B**) Splicing pattern of *COASY* gene in the KY011 tumor. (**C**) Splicing products for *SERPINA1* and *SENP6* in the KY013 tumor. For *SERPINA1*, the arrows indicate the unspliced product and asterisk marks the spliced product. *RPL13a* (lower panel) is used as the internal control. (**D**) Splicing pattern for the *APOPT1* gene in the KY018 tumor. The two arrows show unspliced/spliced products. The asterisk indicates a non-specific band. (**E**) Splicing products for *EPB41L2* (upper panel) and *FBLN2* (lower panel). In breast cancer (KY013, upper panel), the unspliced product of *EPB41L2* is increased in the PDX tumor compared with primary tumor. In colon cancer (KY018), both of the spliced/unspliced transcript level were overall decreased.

## Data Availability

The RNA-seq data are available as follows: Accession to cite these SRA data: PRJNA1039886; Release date: 12 February 2024. The data will be accessible with the following link after the indicated release date: https://www.ncbi.nlm.nih.gov/sra/PRJNA1039886 (accessed on 5 September 2023).

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
