# Peer review of "Comparative RNA-Seq Analysis Revealed Tissue-Specific Splicing Variations during the Generation of the PDX Model"

_ijms, 2023, doi:10.3390/ijms242317001_

Round 1

Reviewer 1 Report

Comments and Suggestions for Authors

The article by Eun Ji Lee and colleagues aims to study tissue-specific gene expression and the regulation of alternative splicing in PDX tumors compared to the primary tumors. As a relatively new strategy for studying tumors, PDX is still far from being fully defined and molecular process studies are needed. Therefore, the study is potentially interesting, however, the manuscripts have serious weaknesses. It is clear that RNA-seq experiments show some different gene expression patterns between different tumor types and between PDX and primary tumor, but it is not sufficiently thorough. The identification of several alternative splicing events is barely confirmed, with only 4 events and not in all tumor types. The authors should support their results with a more accurate demonstration of these differences. 

At the moment the manuscript requires many experiments and I suggest a rejection, the article could be extensively improved and resubmitted later.

Figures and tables need to be improved extensively.

Author Response

We are really thankful for the valuable comments from the reviewer #1. Please check the below for the point-by-point responses for the comments.

Reviewers #1

  • The article by Eun Ji Lee and colleagues aims to study tissue-specific gene expression and the regulation of alternative splicing in PDX tumors compared to the primary tumors. As a relatively new strategy for studying tumors, PDX is still far from being fully defined and molecular process studies are needed. Therefore, the study is potentially interesting, however, the manuscripts have serious weaknesses. It is clear that RNA-seq experiments show some different gene expression patterns between different tumor types and between PDX and primary tumor, but it is not sufficiently thorough.

>> We agree that the limitation of this study is small number of primary-PDX tumor pairs. This is due to the limited tissue availability and manpower. However, for the four pairs, we could see meaningful changes in transcriptomics in tumor type specific manner as shown in the supplementary Table 1-4 as well as in the Figure 3 and Supplementary Figure 2. As the reviewer mentioned, this result is not through but provide a hint for the tumor-specific molecular changes in the formation of PDX.

  • The identification of several alternative splicing events is barely confirmed, with only 4 events and not in all tumor types. The authors should support their results with a more accurate demonstration of these differences.

>> We appreciate for the critical comment and regret that our data for the confirmation of splicing changes was insufficient. To follow the reviewer’s comment, we performed validation experiment for 4 more splicing changes identified from each pri-PDX tumor pairs. The results are shown in the revised figure 6. Now we have validation data for all the four tumor types and found interesting splicing changes for SLK (KY009), COASY (KY011), SENP6 (KY013) and APOPT1 (KY018).

  • At the moment the manuscript requires many experiments and I suggest a rejection, the article could be extensively improved and resubmitted later.

  • Figures and tables need to be improved extensively.

>> We appreciate for the helpful comment. We revised Figure 1, moved some images to supplementary figure and changed fonts of all labels in the tables to increase visibility. For Figure 2-5, we enlarged titles of X, Y axis and added title for each panel to improve visibility and readability. For Figure 6, we added more data with detailed labels to help readers to understand our results. We also revised supplementary tables including supplementary Table 10, with more information and improved uniformity.

Reviewer 2 Report

Comments and Suggestions for Authors

Lee and co-authors assessed whether the splicing pattern of human patient tumor samples would be preserved following their implantation into mice to generate PDX tumor models. They found that significant changes could be observed following the adaptation of the tumor cells to the new environment but that overall, primary and implanted tumors correlated well in regard to tissue-specific genes.

The image quality of the figures, which is quite poor in terms of resolution and clarity, makes analysis of the data itself challenging. This needs to be improved for publication and it would help peer-review if figures would be clearer. The small sample size further dampens my enthusiasm for the study.

Pictures of animals and tumors could be moved to supplementary figures.

Figure 1 is partially labeled in Korean - it would be preferable if all figures are accessible for English language speakers or at least explained in the legend.

Figures 2-5 cannot be assessed in its current form as the labels are too small and the resolution is insufficient. Images in the Supplementary suffer from the same issue - if the axis label cannot be read, the figures are meaningless.

Sequencing data should be deposited in a public database.

Author Response

We are very thankful for the valuable comments from the reviewer #2. Please check the following point-by-point responses to the comments

Reviewers #2

  • Lee and co-authors assessed whether the splicing pattern of human patient tumor samples would be preserved following their implantation into mice to generate PDX tumor models. They found that significant changes could be observed following the adaptation of the tumor cells to the new environment but that overall, primary and implanted tumors correlated well in regard to tissue-specific genes.
  • The image quality of the figures, which is quite poor in terms of resolution and clarity, makes analysis of the data itself challenging. This needs to be improved for publication and it would help peer-review if figures would be clearer.

>> We agree with the reviewer’s comments and think the resolution of images was decreased during the step of copying original figures into the WORD template of IJMS. We revised all figures to improved resolution and also separately provide PDF version of our figures for the evaluation.

  • The small sample size further dampens my enthusiasm for the study.

>> We agree that the limitation of this study is small number of primary-PDX tumor pairs. This is due to the limited tissue availability and manpower. However, for the four pairs, we could see meaningful changes in transcriptomics in tumor type specific manner as shown in the supplementary Table 1-4 as well as in the Figure 3 and Supplementary Figure 2. As the reviewer mentioned, this result is not through but provide a hint for the tumor-specific molecular changes in the formation of PDX.

  • Pictures of animals and tumors could be moved to supplementary figures.

>> We appreciate for the helpful comment. Following it, we moved pictures of animals ans tumors into Supplementary Figure 1.

  • Figure 1 is partially labeled in Korean - it would be preferable if all figures are accessible for English language speakers or at least explained in the legend.

>> We regret that label was not checked carefully and admit this is our mistake. We changed labels in the figure 1 to English. We really appreciate for the valuable comment.

  • Figures 2-5 cannot be assessed in its current form as the labels are too small and the resolution is insufficient. Images in the Supplementary suffer from the same issue - if the axis label cannot be read, the figures are meaningless.

>> We deeply appreciate for the helpful comment. Due to the limitation of the space, It was not easy for some figures to be read for each gene names or pathway names. To improve this limitation, we provided gene list or pathway list in the supplementary Tables separately. For other panels, also carefully checked labels and resolutions in the main figures and supplementary figures to improve readability. For resolution, please check our PDF file for the figures if the revised version is not visible well for some reason.

  • Sequencing data should be deposited in a public database.

>> We are thankful for the critical comment and agree that the sequencing data should be deposited in the public database. Following the comment, we deposited the RNA-seq data into SRA database and received the following confirmation letter.

This is an automatic acknowledgment that your recent submission to the SRA database has been successfully processed and will be released on the date specified.
Please reference PRJNA1039886 in your publication. This BioProject accession number is provided instead of SRP and should be used in your publication as it will allow better searching in Entrez.
Accession to cite for these SRA data: PRJNA1039886
Temporary Submission ID: SUB13968034
Release date: 2024-12-02
Your SRA records will be accessible with the following link after the indicated release date:
https://www.ncbi.nlm.nih.gov/sra/PRJNA1039886

We revised data available section accordingly, with these accession number and link provided.
